# A Link between Chronic Kidney Disease and Gut Microbiota in Immunological and Nutritional Aspects

**DOI:** 10.3390/nu13103637

**Published:** 2021-10-17

**Authors:** Paulina Mertowska, Sebastian Mertowski, Julia Wojnicka, Izabela Korona-Głowniak, Ewelina Grywalska, Anna Błażewicz, Wojciech Załuska

**Affiliations:** 1Department of Experimental Immunology, Medical University of Lublin, 4a Chodzki Street, 20-093 Lublin, Poland; paulinamertowska@gmail.com (P.M.); mertowskisebastian@gmail.com (S.M.); ewelina.grywalska@umlub.pl (E.G.); 2Department of Pathobiochemistry and Interdisciplinary Applications of Ion Chromatography, Medical University of Lublin, 1 Chodzki Street, 20-093 Lublin, Poland; j_wojnicka@onet.eu (J.W.); anna.blazewicz@umlub.pl (A.B.); 3Department of Pharmaceutical Microbiology, Medical University of Lublin, 1 Chodzki Street, 20-093 Lublin, Poland; 4Department of Nephrology, Medical University of Lublin, 8 Jaczewskiego Street, 20-954 Lublin, Poland; wojciech.zaluska@umlub.pl

**Keywords:** gut microbiota, chronic kidney disease, diet, nutrition

## Abstract

Chronic kidney disease (CKD) is generally progressive and irreversible, structural or functional renal impairment for 3 or more months affecting multiple metabolic pathways. Recently, the composition, dynamics, and stability of a patient’s microbiota has been noted to play a significant role during disease onset or progression. Increasing urea concentration during CKD can lead to an acceleration of the process of kidney injury leading to alterations in the intestinal microbiota that can increase the production of gut-derived toxins and alter the intestinal epithelial barrier. A detailed analysis of the relationship between the role of intestinal microbiota and the development of inflammation within the symbiotic and dysbiotic intestinal microbiota showed significant changes in kidney dysfunction. Several recent studies have determined that dietary factors can significantly influence the activation of immune cells and their mediators. Moreover, dietary changes can profoundly affect the balance of gut microbiota. The aim of this review is to present the importance and factors influencing the differentiation of the human microbiota in the progression of kidney diseases, such as CKD, IgA nephropathy, idiopatic nephropathy, and diabetic kidney disease, with particular emphasis on the role of the immune system. Moreover, the effects of nutrients, bioactive compounds on the immune system in development of chronic kidney disease were reviewed.

## 1. Introduction

Social development, economic and socio-geographical factors contribute to the observation of an increased number of kidney diseases. Currently, in addition to independent kidney disease entities, such as chronic kidney disease (CKD) and acute kidney injury (AKI), the participation of these organs in complications of other diseases is noted. The importance of the kidneys for the proper functioning of the human body does not need to be mentioned [1]. The kidneys play a key role in such processes as removing waste products from the blood and maintaining the correct concentration of electrolytes and water in the body. Depending on the body weight, 4 to 6 L of blood can circulate in the human body, which means that every day around 1500 L of blood can flow through the kidneys, being purified by means of nearly a million small filters in the form of nephrons [2]. Despite performing such important functions, the kidneys are one of the most neglected organs. It is related not only to the lack of adequate knowledge and actions in the field of prophylaxis, but also due to the fact that most kidney diseases in the initial stages are asymptomatic. As a result, patients see doctors too late, and the kidney dysfunction is so great that it significantly affects the functioning of other organs in the body [3]. Literature data estimate that CKD occurs in 1 in 10 inhabitants of the globe, while in Poland this problem may affect about 4 million people [4]. The diagnosis of kidney disease is also extremely limited, as in most cases there are no sensitive and specific molecular markers indicative of the development of a particular disease entity. Therefore, new methods and diagnostic tools are sought increasingly often, are aimed at discovering new disease markers that will not only allow for more accurate and early diagnosis, but also to predict risk, increase prognosis and select the appropriate personalized treatment [5]. One of the factors with increasing diagnostic potential is the analysis of the composition of the human microbiota. Understanding the composition, dynamics, and stability of a patient’s microbiota in different areas of the body and identifying changes that occur during disease onset or progression can aid in the development of personalized microbiota-based therapies. Although the literature abound with reports on the role of the human microbiota in the progression of diseases such as obesity, diabetes, and cancer, their importance in the development of kidney diseases is a subject that has not been explored so far [4,6].

The aim of this study is to present the importance and factors influencing the differentiation of the human microbiota in the progression of kidney diseases, such as CKD, IgA nephropathy, idiopathic nephropathy, and diabetic kidney disease, with particular emphasis on the role of the immune system. Moreover, evaluation of the effects of nutrients, bioactive compounds and both conventional and functional foods on the immune system in development of chronic kidney disease were reviewed.

## 2. The Importance of the Human Microbiota

The definition of the human microbiota covers all the microorganisms inhabiting the human body, which consists of three main domains of life: bacteria, archea and eukaryotes. The development of molecular analysis techniques, including genomics and proteomics, has shown that each person has his own unique microbiota pattern in terms of quantitative and qualitative composition, which plays an important role in maintaining health and the occurrence of diseases [7,8,9]. With age, the composition of the basic human microbiota, including firmicutes (60% of the total intestinal microbiota), bacteroidetes (15% of the total intestinal microbiota), actinobacteria and proteobacteria, changes. The stages of human life and the physiological changes occurring during them, as well as environmental factors like ethnicity and geographic location are strongly correlated with the diversity of the human gut microbiota (Figure 1) [10,11,12,13,14].

Under the conditions of homeostasis, the intestinal microbiome performs a number of important functions aimed at supporting the human body, especially in terms of supplementing metabolic dysfunctions of the digestive system. Commensal intestinal microbiota acting as symbionts are responsible for such processes as digestion of complex carbohydrates, vitamin synthesis, maintaining the intestinal epithelium, protection against infections by pathogenic microorganisms, and immune regulation (Figure 2) [15]. With the proper functioning of the organism (understood as the absence of pathogenic symptoms), intestinal microorganisms form communities called enterotypes, which have a completely different effect on the intestine. It should be noted that the enterotypes possessed by a given person are not constant and are subject to dynamic changes conditioned by a number of factors such as diet, lifestyle or environmental stress (Table 1) [16,17,18].

All quantitative and qualitative changes occurring in the gut microbiota are termed dysbiosis and lead to cellular and metabolic disorders that affect the emergence or progression of disease states. The most common cause of dysbiosis is the development of allergies, asthma, diabetes, obesity and kidney disease (Figure 2). In the case of the last group, the causes of intestinal dysbiosis may be iatrogenic factors or uremia, which cause renal dysfunction. Reduction or loss of the filtration capacity of the kidneys causes the secretion of urea into the gastrointestinal tract, which, due to the enzyme urease produced by some microorganisms, undergoes hydrolysis and produces large amounts of ammonia. The presence of ammonia significantly influences the development of the commensal bacteria living in the human intestines and thus the quantitative and qualitative disturbance of the microbiota. Literature data indicate that other factors are also involved in the process of intestinal dysbiosis, such as consumption of medications (antibiotics, orally administered iron), changes in diet (lowering the amount of consumed diet, vitamin K deficiency), metabolic changes (metabolic acidosis, slowing down the passage of the intestinal epithelium) [21,22,23].

## 3. The Process of Immune Modulation by Human Gut Microbiota

Increasingly frequent scientific and experimental studies indicate the important role of intestinal microbiota not only in maintaining the proper homeostasis of the human body, but also in modulating the immune system. In the literature, it can be found that the intestinal microbiota is compared to a separate organ in the human body, the metabolic capacity of which exceeds the range of biochemical reactions taking place in the liver [24]. The importance of the intestinal microbiota is also demonstrated by its quantity. Research shows that people who eat a typical Western diet have 10^10^–10^11^ cfu/g, which in terms of weight means that the human caecum and colon are inhabited by 250 to 750 g of bacteria. They take into account that bacterial biomass may constitute from 40 to 55% of the solid mass of feces, which means that an average person excretes approximately 15 g of bacterial mass per day. A detailed analysis showed that almost 50% of the bacteria excreted despite aerobic conditions (the intestine is anaerobic and the presence of oxygen negatively affects the survival of some species of bacteria) are still alive [24,25,26].

### 3.1. The Role and Importance of Bacterial Metabolites and Components in the Human Body

#### 3.1.1. The Role of Short-Chain Fatty Acids Which Are Products of Intestinal Bacterial Metabolism in the Human Body

Due to the anaerobic conditions in the distal gastrointestinal tract, most biochemical reactions occurring there are based on the fermentation process, which is responsible for the hydrolysis of nutrients in the diet. This process mainly affects polysaccharides, oligosaccharides and disaccharides, which are broken down into simple sugars, which are easily digestible energy compounds for microorganisms. The process of the fermentation of carbohydrate compounds itself leads to the production of short-chain fatty acids (SCFA), H_2_ and CO_2_, while in the case of amino acids and proteins, branched fatty acids are formed [27]. In terms of chemicals, SCFA includes organic acids composed of 1 to 6 carbon atoms in an aliphatic chain, which means that this group includes compounds such as acetic, propionic, butyric, valeric or caproic acid [28,29]. Additional studies have shown that the molar ratios of acetate, propionate and butyrate are also variable. In the colon, this ratio is 60:25:15, respectively, and it varies in individual sections of the intestine depending on factors such as diet, age and disease. The importance of SCFA in the human body may be proven by the fact that 95% of these compounds are absorbed by intestinal epithelial cells, and only 5% is excreted from the body with the feces [30].

The intestinal microbiota, using a number of metabolic processes, are responsible for the production of SCFA in the human body, which occurs at three sites:In colonic epithelial cells where butyrate is the main substrate (which is the energy source for colonocytes);In liver cells, where the acetate produced in the gluconeogenesis process is metabolized, as well as butyrate and propionate; andIn muscles, where the process of generating energy takes place due to the oxidation of acetate [31,32,33].

SCFAs also play a very important role in the protection of the human body. This applies to two aspects: the first is the inhibition of histone deacetylase (HDAC) activity, and the second is the participation in signaling by the complex of free fatty acid receptors coupled with G proteins (GPRs) [34,35].

Histone deacetylase is the enzyme responsible for removing the acetyl group from the ε-N-acetyl lysine found in histone. Such a process enables better wrapping of histones by DNA, which influences gene expression (only hyperacetylated chromatin is transcriptionally active). Literature data indicate that inhibition of HDAC by SCFA depends on many factors, including the type of acid and the type of cells and tissues in which this process occurs [36]. One of the most potent HDAC inhibitors is butyric acid, which, although produced in smaller amounts, plays the most important role in regulating this process. The next two places are propionic acid and acetic acid. There are two mechanisms of HDAC activity inhibition: direct (through the binding of two butyrate molecules to the enzyme pocket) and indirect (through GPR41, GPR43 and GPR109 receptors) [37,38]. Studies have shown that inhibition of HDAC activity by SCFA takes place in all cells of the immune system, both innate and acquired.

The second protective aspect concerns signal transmission by the G protein-coupled free fatty acid receptor complexes. We distinguish two types of complexes: free fatty acid receptor 2 coupled with G proteins—FFAR2/GPR43, which is responsible for the binding of acetates, butyrate, valerate and caproate; and receptor free fatty acids 3 coupled to G proteins—FFAR3/GPR41, which has an affinity for acetate and propionate and little for butyrate, valerate and caproate. The first type of receptor can be found in almost the entire digestive tract (secretory cells in the ileum, colon, colonocytes and enterocytes in the small and large intestine) and cells of the immune (on eosinophils, basophils, neutrophils, monocytes, dendritic cells and mast cells), and nervous systems. Studies have shown that by inducing the secretion of the YY peptide (PYY) and the glucagon-like peptide-1 (GLP-1), SCFAs can influence weight change and reduce the amount of food consumed by humans [39,40,41]. The second type of receptor, is expressed, inter alia, in adipose tissue and in the peripheral nervous system. Activation of GPR41 by SCFA improves glucose tolerance by inducing intestinal gluconeogenesis. Additionally, their presence has been demonstrated in the pancreas on Langerhans cells, in the spleen, and on peripheral blood mononuclear cells (PBMC), but their role in these organs has not been described to date [42,43,44].

By analyzing the effect of SCFAs on the human immune system, it was shown that they are involved in the process of maintaining the balance of anti-inflammatory and pro-inflammatory responses. Thanks to this, SCFAs become a kind of communication channel between the naturally occurring commensal intestinal microbiota and the immune system itself. Numerous studies have shown that SCFAs are directly involved in the differentiation of IL-17, IFN-γ and IL-10-secreting T cells through HDAC inhibition and are indirectly dependent on GPR41 and GPR43 receptors. As a result, these compounds can stimulate the process of T cell differentiation into effector and regulatory cells and can participate in the regulation of pro-inflammatory and anti-inflammatory responses [45,46].

#### 3.1.2. The Role and Importance of Indole

Eating tryptophan-rich foods has a significant effect on intestinal microorganisms. This aromatic amino acid is broken down by bacterial tryptophanase (systentized by many intestinal bacteria, including E. coli) to indole. The concentration of this compound in the human colon is not fully known. Studies show that E. coli strains (both commensal and pathogenic) produce about 500 μM of indole [47] in laboratory conditions. In addition, studies conducted by Karlin et al. and Zuccato et al. have indicated that the concentration of indole in the human stool may vary from 250 to 1000 μM [48,49]. As for the functions of indole in the human body, it is a compound responsible for intercellular signaling, involved in processes such as increasing the expression of genes of intestinal epithelial cell connections or pro and anti-inflammatory factors in intestinal epithelial cells. This makes this compound responsible for maintaining host–microbiota homeostasis on the mucosa surface [50,51]. It should be mentioned that the produced indole is absorbed into the blood from the intestine and is metabolized to indoxyl sulphate in the liver; its residues are excreted in the urine in the case of properly functioning kidneys. This means that the production of indole by the gut microbiota and its uptake by the host cells suggests that there may be an indole concentration gradient in the gut. Excessive production of this compound by bacteria and its conversion into uremic toxin (which may occur due to the individual specificity of the composition of the intestinal microflora or as a result of dysbiosis) may result in impaired proper functioning of the kidneys [52]. Over 600 different compounds belonging to the indole group have been detected in the human body, of which indole acetic acid (IAA) seems to be extremely important. Studies have shown that this compound is only partially removed by hemodialysis in patients with CKD, and that its accumulation in the patient’s body leads to glomerular sclerosis and interstitial fibrosis, which may lead to the progression of CKD [53].

#### 3.1.3. The Role and Importance of Aryl Hydrocarbon Receptor

The aryl hydrocarbon receptor (AhR) has been discovered to mediate toxic reactions induced by halogenated aromatic hydrocarbons and polycyclic aromatic hydrocarbons (e.g., such as 2,3,7,8-tetrachlorodibenzo-p-dioxin (TCDD)) [54]. The inactive form of these receptors is located in the cytoplasm as a complex with chaperones such as HSP90, P23 and XAP2. The ligands for this type of receptor are numerous compounds of both endo- (lipoxin A4, bilirubin and lipopolysaccharides) and exogenous origin, including dietary components, the host metabolism, the intestinal microbiome (mainly derived from tryptophan metabolism) or compounds of environmental origin that correspond to induce AhR conformational changes. These receptors play extremely important functions in the human body, including inducing the expression of genes of pro-inflammatory factors, the metabolism of xenobiotics (CYP1A1, CYP1A2, CYP1B1 and COX-2), or inducing selective protein degradation. Literature data also showed that AhR receptors are also correlated with CKD. It turns out that many uremic toxins which are products of the metabolism of the intestinal microflora have been classified as AhR antagonists. A recent study found that AhR activation in patients with CKD stages 3 to 5 correlates strongly with eGFR and IS levels, and the expression of AHR target genes in the blood (CYP1A1 and AhRR) was found to be increased in patients with CKD, compared with healthy controls [55,56].

#### 3.1.4. The Role and Importance of Polyamines

Another group of compounds are polyamines, which include spermine (involved in cellular metabolism and a growth factor for some intestinal bacteria), putrescine (resulting from the breakdown of proteins by anaerobic bacteria), as well as polyamine oxidase and acrolein. Studies in animal models have shown that these compounds are involved in the development of CKD. Changes in polyamine metabolism resulted from changes in the metabolism of intestinal microorganisms, which resulted in the development of intestinal dysbiosis and thus intensified the progression of CKD. A study in patients diagnosed with CKD showed a decrease in spermine and an increase in plasma putrescine, polyamine oxidase and acrolein, which may suggest that these compounds may act similarly to uremic toxins. To date, scientific and clinical studies have shown that blood creatinine levels are effectively used as a marker for CKD. However, creatinine is not a toxic compound, and a number of studies on the concentration of acrolein (which is a toxin) correlate with kidney damage, and more specifically with the acceleration of the process of kidney fibrosis. Therefore, researchers have postulated using acrolein together with the determination of creatinine levels as a new diagnostic marker of CKD progression [57].

### 3.2. Regulation of the Immune Response by Gut Microbiota

In the intestinal environment, the intestinal epithelium performs the main nutritional and protective functions. It is responsible for the absorption of nutrients and as a protective barrier that often prevents pathogens and antigens from entering. The intestinal epithelium, composed of single layers of cylindrical cells tightly connected with each other, separates the intestinal lumen from the lamina propria, constituting a kind of seal. Consortia of commensal microorganisms found in the gastrointestinal tract are responsible for establishing and/or maintaining homeostasis in the intestinal environment through the process of immunomodulation and the development of a number of mechanisms responsible for maintaining the functional integrity of the intestine [58]. Such mechanisms include participation in the maintenance of the structure of tight junction proteins (claudins, occludins, junctional adhesion molecules (JAMs, belonging to the immunoglobulin subfamily) and tricellulins), the induction of epithelial heat shock proteins, i ncreasing the expression of mucin genes, secretion of antimicrobial peptides and competition with pathogenic bacteria. This means that the gut microbiota is involved in many functions in the field of metabolic (the ability of flora to break down undigested food debris by fermentation, SCFA), trophic (competitive inhibition for biotope and nutrients, and prevention of harmful colonization and multiplication of pathogenic bacteria), and immunological activity [58,59]. The implementation of the latter group of functions is related to the process of eliminating harmful antigens by molecular means by combining TLR receptors and NOD domains (nucleotide oligomerisation domain) with the structures of bacterial cells such as lipopolysaccharide or teichoic acid, which will lead to the induction of the signaling cascade responsible for the secretion of inflammatory mediators. In addition, microorganisms promote cell survival via the phosphatidylinositol 3-kinase or kinase B via the MyD88 factor, which allows the building of a protective barrier against damage caused by stress factors. It has also been shown that the signaling process of intestinal microorganisms by TLRs found in the intestinal mucosa is required to maintain not only the homeostasis of the entire epithelium, but also its repair [60,61].

Detailed analysis of the relationship between the role of intestinal microorganisms and the development of inflammation within the symbiotic and dysbiotic intestinal microbiota showed a number of significant changes. Under conditions of a symbiotic gut microbiota, we can see an increase in commensal bacteria and the maintenance of gut epithelial integrity. The first line of defense is the mucus layer, which is composed of two integral parts: the outer (rich in antibacterial peptides produced by Paneth cells and immunoglobulin A, synthesized by plasma cells) and the inner (responsible for hydration, regeneration processes and protection against the action of digestive enzymes of epithelial cells). This keeps microorganisms away from intestinal epithelial cells, which leads to increased tolerance of the immune system to the commensal microorganisms residing there [62,63]. When the mucous protective layer is compromised, the intestinal epithelial cells use signaling cascades using TLRs to detect microbes. In the case of Gram-negative bacteria, the signaling molecule will be LPS, which will be taken up by TLR4, while in Gram-positive bacteria, teichoic acids will be taken up by TLR2. Upon ligation of the signal molecule to the appropriate TLR, MyD88 is recruited, which activates the NFκβ pathway and leads to the production of antimicrobial proteins and pro-inflammatory cytokines. Under normal microbiota conditions, intestinal epithelial cells are desensitized by continuous exposure to the same LPS derived from commensal bacteria or may be weakened [64,65]. There are three mechanisms involved in this process. The first one concerns the downregulation of IL-1 receptor related kinase 1 (IRAK1), which acts as an activator of the NF-κβ cascade. The second involves the induction of the G receptor, activated by the proliferators of PPAR peroxisomes (which are transcription factors that regulate the expression of genes related to carbohydrate, fat and protein metabolism, as well as cell proliferation and inflammation), which can divert NF-κβ from the nucleus. The third mechanism is based on the inhibition of polyubiquitilation and degradation of I κβ (nuclear factor kappa β inhibitor), which inactivates NF- κβ. Exposure to LPS or teichoic acid induces epithelial cells to secrete TGF-β (transforming growth factor beta), BAFF (TNF family B-cell activating factor) and APRIL (proliferation-inducing ligand), which are responsible for the development of immune cells tolerating the inhabiting microbiota. This process also involves dendritic cells that support the development of IL-10 and TGF-β secreting Tregs and stimulates the production of an IgA specific for commensal microorganisms [66,67] (Figure 3A).

In the case of dysbiosis of the intestinal microbiota, the number of commensal microorganisms is reduced in favor of pathogenic microorganisms, the accumulation of toxins (mainly urea and ammonia), and the integrity of the intestinal epithelium is disrupted, which in turn leads to the development of inflammation. When the continuity of the intestinal epithelium is interrupted, bacteriaand bacterial cell components are translocated. This leads to a situation where the gut’s immune system directs a pro-inflammatory response to remove pathogenic bacteria. This is possible through the secretion of IL-1 and IL-6 from intestinal epithelial cells, by the promotion of the Th1 and Th2 responses by dendritic cells and macrophages, and by the production of higher levels of specific IgG by B cells. When a bacterial factor such as LPS binds to the receptor complex (CD14-MD2-TLR4) within macrophages, it results in the activation of the signaling cascade with the activation of p38 ^MAPK^ (mitogen-activated protein kinases), leading to the production of significant amounts of inflammatory cytokines, such as INF -β, INF-γ, IL-1b, IL-6, TNF-α and IL-12. The presence of toxic compounds called uremic toxins is one of the causes of inflammation affecting the development of kidney disease [68,69] (Figure 3B).

## 4. Influence of the Intestinal Microbiota on the Development of Kidney Diseases, with Particular Emphasis on the Role of the Immune System

Any abnormalities of the intestinal microbiota resulting in its dysregulation may lead to inflammation and thus cause a number of diseases, including kidney dysfunction. Currently, the changes observed in the intestinal microbiota affecting renal dysfunction include: Reduced diversity and number of microorganisms, with a predominance of proteolytic bacteria;A phenomenon of the translocation of the microorganisms associated with the colonization of regions of the gastrointestinal tract that have been much less populated so far, and changes in the ratio of aerobic and anaerobic bacteria;The intestinal epithelial barrier is disrupted; andThe production of uremic toxins [70,71].

In many cases of people with kidney diseases, including extreme cases of kidney failure, the synthesis of many toxic compounds in the human body is accelerated, which leads to an increase in the concentration of uremic toxins in the plasma and the progression of kidney diseases. A normal gut microbiota produces compounds that are normally excreted by the kidneys, but which can also be considered potentially toxic. This is the case with the bacterial fermentation of the tyrosine amino acids, which are obtained from the diet through the consumption of meat and dairy products, to a compound in the form of p-cresol. This is also the case with the fermentation of tryptophan to indole. After absorption in the colon, these compounds are metabolized in the liver and converted into the toxic forms of p-cresyl sulphate and p-indoxyl sulphate [70,71,72,73]. Both of these compounds have an affinity for albumin, which means that they can exist in the human body in two forms: the free fraction and the serum-bound fraction. The third type of harmful compounds formed are amines. More precisely, they are compounds of choline and phosphatidylcholine fermentation by intestinal bacteria to trimethylamine, which are converted in the liver into trimethylamine N-oxide (TMAO) (Table 2). All of these toxins are mainly eliminated from the body by the kidneys, and more specifically by the kidney tubules, and an excessive quantity of these toxins leads to damage to the kidney function. As a result of breaking the continuity of the intestinal barrier, these compounds may also enter the systemic circulation and influence the development of cardiovascular diseases as well as symptoms of the central nervous system [74,75,76,77].

### 4.1. Chronic Kidney Disease

The importance of gut microbiota in the progression of chronic kidney disease requires an understanding of many interrelated aspects, including the composition, dynamics, stability, and interactions between bacteria and the human body. The development of CKD is largely related to the accumulation of uremic toxins like indoxyl sulfate, p-cresol sulphate and trimethylamine N-oxide.

The first toxin is p-indoxyl sulphate, which is an indole derivative produced by metabolism in the liver. This compound is a ligand for the acrylic hydrocarbon receptor and acts as a transcriptional regulator. In patients with CKD, this compound is not excreted in the urine and accumulates in the body. Literature data indicate that the elimination of this toxin from the body is extremely difficult, and its reduction ratio with regular hemodialysis is only 31.8% [78]. Scientific studies in animal models have also shown that p-indoxyl sulphate can damage renal tubular cells and mediate changes in the expression of the TGF-β1 gene and the tissue metalloproteinase inhibitor associated with tubulointerstitial fibrosis. In addition, studies conducted by Ichii’s team on mouse models showed the effect of this uremic toxin on the altered pro-inflammatory phenotype of podocytes, which was accompanied by decreased expression of genes specific for these cells, as well as their reduced viability [79]. Moreover, it has also been shown that an excessive amount of indoxyl sulfate also affects the activation of the renin-angiotensin-aldosterone system in the kidneys of mice [79,80].

Another uremic toxin which is a product of the breakdown of the amino acids tyrosine and phenylalanine is p-cresol sulphate produced in the liver. In the case of fully functional kidneys, this compound is excreted in the urine, which depends on tubular secretion by specific transporters. In patients with CKD, these transporters are impaired and the toxin accumulates in the body [81,82]. Literature data estimate that this toxin cannot be effectively removed by dialysis, and the reduction factor is only 29.1% with regular hemodialysis [78]. In animal models, this compound has been shown to lead to the increased expression of numerous transcription factors, such as fibronectin and L-actin in smooth muscle within proximal tubular cells. In addition, experiments in mice with partial nephrectomy showed that p-cresol sulfate was responsible for the activation of the intrarenal renin-angiotensin-aldosterone system, and also led to interstitial fibrosis and glomerulosclerosis [80]. Research data compiled by the Meijers team, which included nearly 500 patients, showed that there is a correlation between the level of p-cresol sulphate and the development of CKD. This mainly concerns the glomerular filtration rate, the values of which decreased with the increase in the concentration of the uremic toxin tested. A similar relationship was found in patients with end-stage renal failure treated with hemodialysis, where it was related to the increased risk of death in patients with increased levels of p-cresol sulphate [81]. The research team led by Lin showed that there is a negative correlation between serum levels of p-indoxyl sulphate and p-cresol and renal function in patients with CKD. Based on the obtained studies, it was found that both of these uremic toxins exert a number of negative effects on numerous cellular processes, such as stimulation of oxidative stress, fibrosis and inflammatory reactions. Additionally, it has been established that high plasma levels of both of these toxins are correlated with progression to end-stage renal disease and increased mortality in patients with CKD [83].

TMAO, a breakdown product of choline, phosphatidylcholine, and dietary L-carnitine, is also considered an extremely important uremic toxin. Studies have shown that there is a positive correlation between the concentration of this compound in the blood and the presence of bacteria from the *Clostridiaceae* and *Peptostreptococcacea* families [84]. Experiments in animal models showed that the increased concentration of this compound significantly correlated with an increase in tubular interstitial fibrosis, collagen deposition and changes in the degree of phosphorylation of Smad3, which is an important regulator of this process. The studies conducted by the Tang team showed a correlation between the concentration of TMAO and the development and progression of CKD also in humans [85]. Their studies showed that elevated levels of this uremic toxin were associated with an almost 70% higher risk of death in CKD patients (even after adjusting for traditional risk factors and CRP protein). Due to all the above-mentioned changes caused by TMAO and the increased mortality of patients with CKD who have excessive levels of this uremic toxin, it is extremely important and warranted to continue further research that may allow for a detailed determination of the role of this compound in the progression of CKD [86]. Additionally, it has been shown that TMAO is involved in changes in cellular metabolism, that is, it affects the metabolism of cholesterol and bile acids, is responsible for the stimulation of the expression of scavenging receptors on macrophages and it modifies the sterol transporter in the liver and intestines [84,85,86,87,88].

A consequence of the accumulation of uremic toxins in CKD is also the very occurrence of the disease, and more specifically its impact on the intestinal microbiota. Patients diagnosed with CKD have poorer diets, especially when it comes to consuming dietary fiber, often use antibiotics and oral iron supplementation, and are at risk of lactic acidosis. In addition, such patients are characterized by slower passage of colon cells or changes in the intestines associated with volume overload with congestion of the intestinal walls, as well as intestinal edema. These changes are reflected in the composition of the intestinal microbiota. Among patients with CKD, a decrease in the number of *Lactobacillus* spp. and *Prevotella* ssp. bacteria, which belong to the normal microbiota of the colon, and nearly 100 times more bacteria from the *Enterobacteriaceae* and *Enterococcaceae* families, the number of which in the normal microbiota is significantly lower, are often found [5,89]. The observed dysbiosis of the gastrointestinal tract also has consequences for the human body itself. This is because the tight connections between the intestinal epithelial barrier and the translocation of bacteria and components of bacterial origin are loosened, which triggers an immune response inducing inflammation. The process of intestinal dysbiosis may also be influenced by other mechanisms related to the increased secretion of urea from the gastrointestinal tract. This means that urea hydrolysable microorganisms produce large amounts of ammonia, to which the commensals of gastrointestinal bacteria is sensitive. The most common causes of the development of chronic kidney disease include the presence and progression of primary and secondary glomeluropathy, diabetic nephropathy and hypertensive nephropathy [90,91].

### 4.2. Idiopathic Nephrotic Syndrome

Idiopathic nephrotic syndrome (INS) is one of the glomerular disorders characterized by edema, proteinuria and hypoalbulinemia. This disease causes gromeluropathy in children from one to 30% of cases in adults, and the pathogens themselves are still the subject of research by many scientists. Research has shown that the immune system is strongly involved in the pathomechanism of INS formation. The underlying mechanism is the disruption of the glomerular permeability barrier, which is caused by the stimulation of antigen presenting cells and B lymphocytes in response to the emergence of allergens or infections. As a result of such stimulation, T lymphocytes are also activated through antigen presentation and cytokine production. Literature data show numerous changes in the T cell population in INS patients. The most frequently observed changes include a decrease in the number of CD4 plus T helper lymphocytes (Th), which is associated with the presence of cytotoxic CD8 plus T lymphocytes, imbalance between Th2 and Th1 lymphocytes, and reduced frequency and function of regulatory T lymphocytes (Tregs), as opposed to increased Th17 cell activity [92].

Additionally, this unit has different histopathological types, including minimal change disease (MCN), membranous nephropathy (MN), focal segmental glomerulosclerosis (FSGS) and membrane proliferative glomerulonephritis (MPGN) [93]. Due to such a wide variety of the disease, researchers have investigated whether intestinal dysbiosis can lead to development (INS) and what potential factors may predispose to the development or progression of this disease. Detailed analysis of the intestinal microbiota at the level of quantitative and qualitative differentiation of taxa carried out by the He team in 2021 [94] showed that there are differences between the microbiota of healthy and INS patients. In patients with INS, fewer bacteria belonging to the following types: Acidobacteria, Firmicutes: especially Negativicutes class, Selenomonadales class, *Veillonellaceae* family, *Clostridiaceae*, and genera: *Dialister, Rombousia, Ruminiclostridium, Lachnospira, Alloprevotella, Closstridium* and *Megamonas* were detected in comparison to healthy individuals. There were also changes in the types of bacteria such as *Parabacteroides* spp., *Bilophila* spp., *Enterococcus* spp., *Eubacterium* spp., which were higher in INS patients than in controls. These observations resulted in the establishment of a clear specific type of bacterial pattern that decreased the amount of bacteria capable of producing SCFA in INS patients. In addition, significant relationships between selected clinical parameters and the presence of individual bacteria were demonstrated. There was a negative correlation between serum creatinine concentration and the occurrence of Burkholderiales, *Barnesiella* spp. or *Alcaligenaceae*, as well as a positive correlation between proteinuria and the occurrence of *Coriobacteria,* Nitrosomonadales, Verrucomicrobia and *Blautia* spp. However, the close interrelationships between clinical parameters and the amount and diversity of the gut microbiota requires much research to fully understand [95,96].

#### Membrane Nephropathy and Mesangial Proliferative Glomerulonephritis

Membrane nephropathy (MN) is also one type of primary glomerulonephritis. It is one of the most common causes of nephrotic syndrome development in adults all over the world. The disease is caused by immunological disorders related to the production of autoantibodies against PLA2R (phospholipase A2 receptor) and HSD7A (Thrombospondin Type 1 Domain Containing 7A) antigens, which belong to the IgG4 immunoglobulin subclass. The second disease is Mesangial proliferative glomerulonephritis (MPGN) which can also manifest as nephrotic syndrome. The most common symptoms are proteinuria, low blood protein levels, high cholesterol, high triglycerides, and edema. A characteristic feature of MPGN is the increased number of mesangial cells in the kidney glomeruli, leading to their damage. Due to the diversity of these diseases and their classification as histopathological subtypes of INS, some researchers have examined whether the development of these disease entities may be influenced by intestinal microorganisms [97,98]. Studies have shown that there are differences between the quantitative and qualitative scales of the gut microbiota of both studied disease entities. Higher amounts of bacteria belonging to Proteobacteria and Gammaproteobacteria have been shown in patients diagnosed with MN than with MPGN. This also concerned changes within the orders of Enterobacteriales, Erysipelotrichales, Enterobacteriaceae, Rikenellaceae, Ruminococcaceae, the Coriobacteriia class, or the *Tyzzerella, Alistipes, Lachnospira, Odoribacter* genera, which were dominant in patients with MN. In the case of MPGN, bacteria from the Rhodobacterales order, *Phyllobacteriaceae* families, and the *Terrimonas* and *Mesorhizobium* genera dominated. The presented differences in taxa may be used in the future as biological diagnostic markers to help distinguish MPGN from MN [94,97].

### 4.3. IgA Nephropathy

IgA nephropathy is one of the most common conditions associated with primary glomerulonephritis. The hallmark of this disease is glomerulonephritis accompanied by the deposition of the IgA1 subclass in these structures. The main IgA receptor is CD89, which is expressed on the surface of monocytes and plays an important role in the pathogenesis of the disease. As shown by the results of studies available in the literature, it can be considered a prognostic marker of disease development. IgAs are produced by Peyer’s patches in the lymphoid tissue of the gastrointestinal mucosa. This area is composed of lymph nodes rich in a large number of B lymphocytes, and between them (in the areas between the clumps) we can find T lymphocytes. The entire structure is covered with a layer of specialized M cells, which are responsible for the uptake of antigens (e.g., bacterial antigens) from the intestinal lumen, which are then transferred to macrophages or dendritic cells involved in the presentation of antigens to T lymphocytes. Due to this fact, under the influence of bacterial antigens of pathogenic origin or commensal Peyer’s patches capable of producing IgA1, an excess quantity is the first step in the development of nephropathy [13,96,99,100]. There is no doubt that the intestinal microbiota may affect the progression of this disease entity. Therefore, many researchers have documented these relationships. The team of De Angelis approached the subject in the most detail, in which the microbiota of IgAN patients were compared with not onlt the control group, but also patients with IgAN progressors and non-progressors. This study involved a very comprehensive approach, with the development of culture-dependent and independent methods, as well as the inclusion of metabolomics analysis. Eight types of bacteria were identified in all studied groups of patients, the dominant of which were representatives of Firmicutes, Bacteroidetes and Proteobacteria, constituting over 98% of all 16S rDNA and 16S rRNA. There were also differences between the 16S rDNA and 16S rRNA analysis, which indicated the types of metabolically active bacteria. In the case of Firmicutes bacteria, their metabolic activity increased in patients diagnosed with IgAN (both subtypes) compared to controls. On the other hand, an inverse relationship was noted for Bacteroidetes bacteria, which were more numerous in patients from the control group. In addition, it was observed that the number of both completely and metabolically active Proteobacteria was lower in healthy subjects than in IgAN patients, while the opposite relationship was found for Actinobacteria, which were more numerous in healthy subjects than those diagnosed with IgAN [96].

Other studies have shown that in patients diagnosed with IgAN there is an increase in the numbers of bacteria from families, such as *Enterobacteriaceae, Ruminococcaceae, Streptoccaceae, Eubacteriaceae*
*and Lachnospiraceae*. There was also a decrease in such genera of bacteria as *Clostridium, Lactobacillus, Enterococcus* and *Bifidobacterium*. Although the latter type of bacteria is commonly considered to be a microorganism with a pro-health potential involved in a number of immunomodulatory mechanisms, inhibiting the development of pathogens or producing SCFA, it should be remembered that with intestinal dysbiosis it may increase its invasive potential. Their amount has been confirmed to be higher than that of healthy people in the case of ulcerative colitis, which may indicate that some species of bacteria are specific for the disease, and further studies on their influence on maintaining normal intestinal homeostasis are necessary [13,101,102,103,104]. Some scientists, similarly to MN and MGPN, showed significant relationships between clinical parameters and the presence of specific microorganisms in the course of IgAN. This concerns the correlation between high levels of albumin and the prevalence of *Prevotella* spp. bacteria, which has also been associated with improved glucose metabolism and insulin sensitivity. The researchers observed a negative correlation between albumin and bacteria of the genera *Klebsiella, Citobacter* and *Fusobacterium*. Additionally, the correlation between the occurrence of *Klebsiella* spp. and increased disintegration of intestinal epithelial cells at IgAN has also been demonstrated [97].

### 4.4. Diabetic Nephropathy

Diabetic nephropathy is one of the most important complications of diabetes worldwide, and as shown by numerous scientific studies, abnormal intestinal microbiota may be involved in its development. Many clinical trials have found an increased level of inflammatory markers in patients with diabetic nephropathy. The cause of inflammation itself is not fully understood, but it is estimated that it may be related to tissue damage, trauma, and patients’ susceptibility to infections. Then the immune cells of T lymphocytes, macrophages, and dendritic cells of both the innate and adaptive systems, as well as other metabolic signals that contribute to disease progression, are activated. Disturbances in the gut microbiota have been observed and documented for both type 1 and type 2 diabetes. Therefore, only detailed studies that can produce evidence of a better understanding of the interaction between the intestinal microbiota and diabetes can help in the development of effective treatments not only for diabetes itself, but also for its complications, such as diabetic nephropathy. For both subtypes of diabetes, there are several interrelationships between microorganisms and disease progression. This applies mainly to disturbances of the intestinal mucosa barrier, which is associated with an increased translocation of bacteria and bacterial components influencing the development of inflammation and insulin resistance. Additionally, studies of fecal transplantation carried out by scientists have shown that changes in the intestinal microbiota directly affect the course of both type 1 and type 2 diabetes [101]. The number and composition of the gut microorganisms also changes drastically during the development of diabetes. There is a decline in the number of bacteria, including *Lactobacillus* spp., *Bifidobacterium* spp., and *Roseburia* spp., which are involved in the immunomodulation process, SCFA production, and the process of supporting the integrity of the intestinal epithelium by producing tight junction proteins. In place of commensal microorganisms, bacteria with high pathogenic potential, such as *Clostridium* spp., *Bacteroides* spp., *Betaproteoovibacter* spp., *Prevotella* spp. or *Desulfovibrio* spp., multiply and increase the permeability of the intestinal mucosa barrier by producing toxins. In the case of type I diabetes, the growth of *Leptotrichia googfellowii*, which has an antigen on its surface that stimulates CD8+ T cells to attack pancreatic islets, has also been observed due to the phenomenon of molecular mimicry, which allows the development of diabetes. In addition, some of the intestinal microbiota (*Lactobacillus* spp., *Bifidobacterium* spp., *Clostridium* spp., *Bacteroides* spp. or butyrate-producing bacteria) may participate in the process of Treg cell differentiation, the number of which is reduced in type 1 and 2 diabetes [98,105].

Additionally, the development and progression of diabetes may be influenced by changes in the endocrine function of the intestines and the composition of metabolites produced by the intestinal microbiota. Butyrate-producing bacteria, as one example of SCFA (*Lactobacillus* spp., *Bifidocacterium* spp.), protect against the development of diabetes by inducing apoptosis in pancreatic islet macrophages. In addition, SCFA has the function of inducing the secretion of GLP-1 (glucagon-like peptide-1), which improves blood glucose levels and reduces insulin resistance in type I diabetes, while stimulating insulin secretion in type 2 diabetes. A complex relationship exists between the intestinal microbiota, the intestinal metabolism, the pathogenesis of diabetes and diabetic nephropathy [98,105,106].

## 5. The Importance of Diet in the Progression of CKD

In patients with CKD, body composition disorders are very common, resulting from excess body fat (leading to obesity) accompanied by muscle wasting. Both these factors not only influence the patients’ problems in everyday existence, but also significantly lower their prognosis. This is largely due to metabolic changes occurring during the course of the disease, i.e., as shown in the literature, to the imbalance in the metabolic balance of insulin-dependent tissues. This means that in the muscles of CKD patients there is an increase in catabolic processes (regulated by glucagon, glucocorticosteroids, catecholamines or pro-inflammatory cytokines), accompanied by an increase in anabolic processes in adipose tissue. Such changes are caused by decreased physical activity of the patients (with decreased muscle strength), the development of metabolic acidosis, and insulin resistance [107,108,109]. The process of disease progression is influenced by the nutritional status of the patient’s body, i.e., the balance between the consumption, absorption and use of nutrients by the body. Literature data estimate that malnutrition is present in approximately 20% of all patients diagnosed with CKD, which strongly correlates with the severity of the disease [110,111]. Therefore, in many regions of the world teams or committees of researchers, doctors and nutritionists are formed to prepare detailed recommendations regarding the diet used during the treatment of CKD, e.g., the National Kidney Foundation (NKF) [112], American Dietetic Association, Academy of Nutrition and Dietetics (ADA) [113], the International Society of Renal Nutrition and Metabolism (ISRNM) [114], and the European Dialysis and Transplant Nurses Association/European Renal Care Association (EDTNA/ERCA) [115]. The recommendations of these teams concern a number of factors, from recommendations regarding the energy values of the diet, through recommendations on the amount of consumed macronutrients (protein, phosphorus, potassium or sodium), vitamins (vitamins C, B12, D, folic acid), to the consumption of minerals. Very often, the selection of the appropriate proportions of the diet depends on the value of the glomerular filtration rate (eGFR), as well as the severity of comorbidities. However, not all countries around the world operate in the same way and include the establishment of teams of specialists to develop such guidelines. Additionally, it should be considered that reports prepared by expert teams from individual countries differ from each other. This is related to the diversity of food products consumed in a given region of the world, as well as their profile, e.g., the dominance of plant-derived products in the diet, but also the availability of specialized food for patients and the scope of financing health services that allow for the modification of the diet composition of CKD patients. Many literature data supported by research indicate that dietary regulation in patients with CKD associated with, among others, a reduction in the consumption of protein, fats, carbohydrates or antioxidants, affects therapeutic success. However, changes in nutrients and the macro- and microelements or minerals they contain may significantly affect the composition and proper functioning of the microbiomes of these patients and there may be accompanying changes in immunomodulatory processes. Thus, it is extremely important to ensure the proper homeostasis of patients’ nutrition and the functioning of their microbiomes in order to increase the prognosis and therapeutic success [111,116].

## 6. How to Restore the Symbiosis of the Gut Microbiota?

Many scientists discuss methods of effectively restoring the symbiosis of the gut microbiota in patients with renal dysfunction as one of the countermeasures in the process of disease progression. However, the studies conducted to date provide only limited evidence and contradictory information regarding the effectiveness of the actions taken by the researchers. The first obvious strategy is to change the diets of such patients. For chronic kidney disease, this diet is rather low in fiber, phosphorus and potassium. This, of course, translates into deficiencies of prebiotic compounds, including consuming the right quantities of dairy products rich in lactic acid bacteria or fiber-rich fruits and vegetables. Several studies concluded that the inclusion of high-fiber foods in the diet decreased the levels of uremic toxins in the body of patients with kidney disease [117,118,119,120]. Another approach is to include probiotics, prebiotics and even symbiotics in the diet (Table 3). Live microorganisms are considered probiotics, which, in an appropriately selected amount, have a positive effect on human health, this applies mainly to lactic acid bacteria, but also to some strains of yeast or mold. These microorganisms are characterized by the ability to colonize various environments in the human body, in particular the intestines, where they play a key role in stimulating the passage of intestinal epithelial cells and ensure the proper development of commensal microorganisms. Prebiotics are food ingredients that selectively influence the development of a specific group or type of microorganism with probiotic properties in the gastrointestinal tract [121,122,123].

Prebiotics can occur naturally in many plants or artificially as food additives or pharmaceutical preparations. One feature of prebiotics is the fact that they are not digested by enzymes in the human body and can only be used by specific microorganisms equipped with an enzymatic apparatus for their decomposition. The last group are symbiotic, i.e., a combination of pro and prebiotics, which together show a synergistic effect influencing the development of normal intestinal microbiota. In addition, it has been shown that they can participate in reducing the concentration of undesirable toxins or metabolites in the human body, and are also involved in the processes of preventing putrefactive reactions in the intestines and the formation of constipation or diarrhea [121,122,123].

Research with the use of probiotics, prebiotics or their combination in symbiotics for the progression of kidney diseases is relatively innovative. Several studies report that the use of probiotics reduced the concentration of uremic toxins, especially p-cresol sulphate and p-indoxyl sulphate in patients with chronic kidney disease and patients undergoing hemodialysis [124]. Research conducted by the Ranganathan team [125] and the Alatriste team [126] showed that the use of probiotics in non-dialyzed CKD patients reduced the level of urea in the serum. In addition, the analysis of individual components of the immune system conducted by the Wang team [127] showed that the use of probiotics for a period of 6 months allowed for the reduction of TNFα, IL-5 and IL-6 levels. In the case of the use of prebiotics, many research teams have shown a decrease in serum and plasma p-cresyl sulphate levels by nearly 20% [128,129,130] and a decrease in serum TMAO in patients diagnosed with CKD [131]. The use of prebiotics additionally affects the immune system by reducing the level of TNFα and IL-6, as was the case with the administration of probiotics [132].

Few studies also mention the use of adsorbent compounds, including AST-120, which is an orally administered carbon adsorbent involved in the removal of uremic toxins [133,134]. It has been shown that this compound adsorbs p-indoxyl sulphate and allows it to reduce the rate of decline in renal function or delay the start of dialysis treatment. However, despite advanced work on animal models, the use of this compound in humans has been approved only in a few Asian countries such as Japan, Korea and the Philippines [135]. Modern medicine is also increasingly searching for new, extremely innovative methods of restoring intestinal symbiosis. This applies to the use of so-called intelligent bacteria, whose genetic modification allows for the delivery of therapeutic agents to the body or the uptake of uremic toxins [24,136,137]. Intestinal microbiota transplant therapy, which has been used to treat chronic Clostridial-induced diarrhea, is also used, and its modifications in animal models allow for good results in restoring the balance of the intestinal microbiota in other diseases [138,139].

In addition to dietary considerations, significant attention should also be paid to the level of physical activity in patients with CKD. As mentioned above, in this group of patients the muscle metabolism is significantly impaired, and there is an excessive accumulation of fat in the body usually caused by restriction of movement or leading a sedentary lifestyle. Due to this fact, many comorbidities develop, such as cardiovascular diseases, hypertension and diabetes. Recent studies have shown that the activation of people with CKD allows them to maintain well-being and improve functional capacity (especially in the context of improving aerobic capacity, kidney function and reducing the risk of other comorbidities) [140].

Due to the increasing number of cases of the development of CKD among people, research has also begun to determine whether the progression of this disease is significantly influenced by other aspects of human life, including not only lifestyle but also addictions. One such example is a study of effect of smoking on the development of CKD. However, the obtained results are not unequivocal. In studies by Yacoub and Habib, smoking has been shown to significantly increase the risk of CKD compared to the control group, especially in patients diagnosed with hypertensive nephropathy. However, research conducted by Xia team based on the meta-analysis of available literature data showed that cigarette smoking is an independent risk factor for CKD. Further comprehensive research is needed to see if quitting smoking can reduce the incidence of CKD in the general adult population [141,142].

## 7. Conclusions

The pathogenesis of chronic kidney diseases involves not only immune dysregulation, but also the genetic susceptibility of people suffering from this type of disease. Additionally, a number of environmental factors are involved in the process of the progression of many kidney diseases, which may directly and indirectly affect the intestinal microbiota and its interactions with the human body. That is why research combining a comprehensive approach to the role of the intestinal microbiota at various stages of the occurrence of kidney disease becomes so important. However, this requires the involvement of genetic, immunological and dietary approaches in determining interactions in the gut-kidney axis. Only by comparing a large number of appropriately selected patients with specific kidney diseases, active and inactive microorganisms inhabiting the intestines and their participation in the immunomodulation process will it be possible not only to understand the prevalence and progression of the disease, but also to develop effective methods of diagnosis and treatment. The research should also consider dietary factors, which, as many studies have shown, have a very significant impact on the differentiation of microorganisms and the development of intestinal dysbiosis. Maintaining the intestinal microbiota in conditions of symbiosis is a challenge for a healthy person, and it is also a necessary aspect in the fight against any type of disease, and the occurrence of specific species, genera or the number ratios of individual bacterial families may in the future become potential diagnostic biomarkers and undoubtedly therapeutic goals that modern medicine is currently facing.

## Figures and Tables

**Figure 1 nutrients-13-03637-f001:**
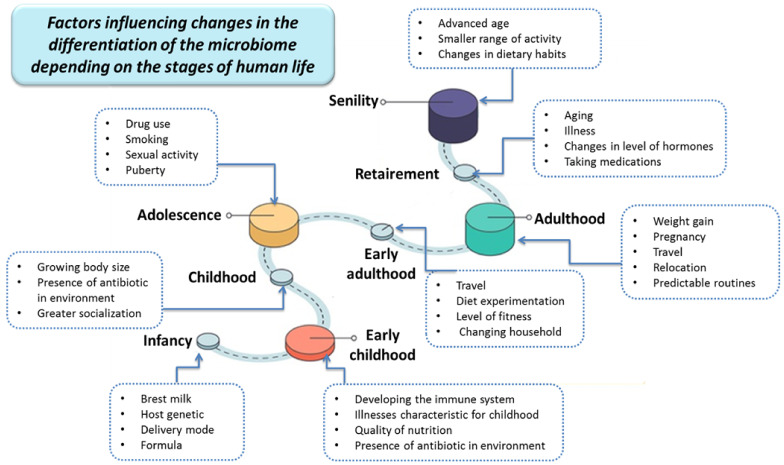
Factors influencing changes in the differentiation of the microbiome depending on the stages of human life (based on [11]).

**Figure 2 nutrients-13-03637-f002:**
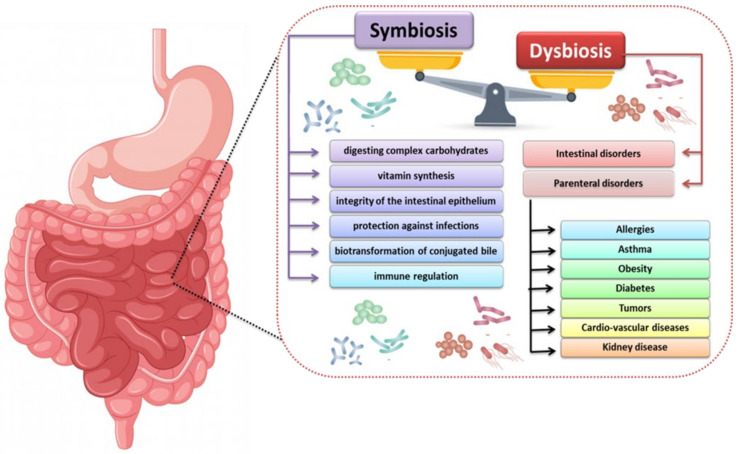
The importance of symbiosis and dysbiosis in the human intestinal microbiota in maintaining homeostasis and pathological changes (based on [16,17,18]).

**Figure 3 nutrients-13-03637-f003:**
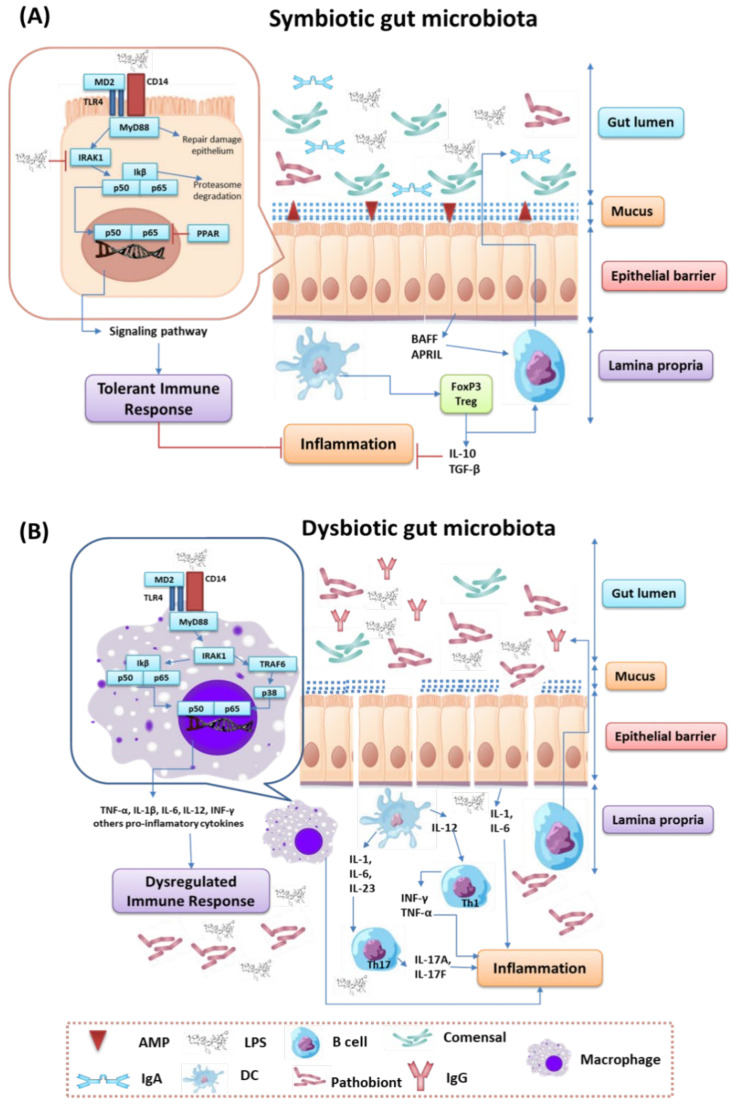
Changes in the symbiotic (**A**) and dysbiotic (**B**) intestinal microbiota in the context of inflammation (based on [64,65,66,67,68,69]).

**Table 1 nutrients-13-03637-t001:** Diversity of enterotypes of intestinal microorganisms depending on the type of microorganisms, energy source, vitamin production capacity and dietary components (based on [16,17,18,19,20]).

Enterotype Name	Microbiological Diversity	The Main Source of Energy	Production of Vitamins	Diet or Diet Components
Enterotype I	The most common bacteria are *Bacteroides* spp.	sugars and protein fermentation	biotin, riboflavin, panthenol, ascorbic acid and thiamine.	meat and products such as mayonnaise, cream, cheese, and other products containing large amounts of saturated fat
Enterotype II	The most common bacteria are *Prevotella* spp.	high ability to break down glycoproteins, especially mucins	biotin, riboflavin, panthenol, ascorbic acid and thiamine.	simple sugars and in vegetarians, Mediterranean ones, rich in fruits and vegetables
Enterotype III	The most common bacteria are *Ruminococcus* and *Akkermansia* spp.	protein fermentation, decomposition of mucin and simple sugars	biotin, riboflavin, panthenol, ascorbic acid and thiamine, folic acid	excess of alcohol and products rich in polyunsaturated fatty acids

**Table 2 nutrients-13-03637-t002:** Formation and health consequences of selected uremic toxins (based on [70,71,72,73,74,75,76,77]).

Compound	Source in the Diet	Compound by Microorganisms Transformation	Compound by Liver Metabolism	Health Consequences
Tyrosine	Turkey, chicken, beef, brown rice, fish, milk, nuts, yogurt, eggs, cheese, fruit and vegetables	p-cresol	p-cresol sulfate	Increased gene expression associated with tubular interstitial fibrosis, aorta and vascular calcification, endothelial cell damage. It lowers the production of erythropoietin and bone rotation
Tryptophan	Beef, poultry, pork, fish, milk, yoghurt, eggs, soy products	Indole	p-indoxyl sulfate	Renal fibrosis, oxidative stress, increased inflammation cytokines, mortality. Braking endothelial proliferation, increased endothelial permeability.
Phosphatidylcholine and choline	fish and seafood, meat and dairy products	Trimethylamine	Trimethylamine N-oxide (TMNO)	Associated with higher mortality
Protein and nitrogen compounds	Dairy products, eggs	Urea	Ammonia	Damage to intestinal epithelial cells due to an increase in the pH of the intestinal environment

**Table 3 nutrients-13-03637-t003:** Comparison of probiotics, prebiotics and symbiotics (based on [121,122,123].

Features	Probiotics	Prebiotics	Synbiotics
Examples of microorganisms	*Lactobacillus* spp., *Streptococcus* spp., *Saccharomyces* spp., *Aspergillus* spp.	-	*Lactobacillus rhamnosus, Bifidobacterium lactis*
Diet ingredients rich in these microorganisms or compounds	They are found mainly in fermented products dairy, pickled vegetables and fruits, fermented sausages, sourdough cakes, sauerkraut, beer, wine and food silage as well as pharmaceutical preparations	Natural: of plant origin, including in garlic (9–16%), chicory (13–20%), artichokes (15–20%), asparagus (10–15%), onions (2–6%), wheat (1–4%) and bananas (0.3–0.7%) Artificial: lactulose, galacto-oligosaccharides, fructo-oligosaccharides, malotoligosaccharides, cyclodextrins, lactosucrose	Pharmaceutical preparations which contain selected strains of bacteria with additives that facilitate colonization, such as inulin or bean fibers, fermented milk drinks

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
