# Peer review of "A Link between Chronic Kidney Disease and Gut Microbiota in Immunological and Nutritional Aspects"

_nutrients, 2021, doi:10.3390/nu13103637_

Round 1
Reviewer 1 Report
This is a very comprehensive review in which Mertowska et al. summarize the present findings on the role and association of gut dysbiosis in different types of CKD.
The authors put emphasis on the role of inflammation and dietary factors in the gut-kidney axis.
Although this is a clear and very well-written review I do have some comments for improving the manuscript.
In paragraph 3, other bacterial metabolites/components should be described beyond SCFA; e.g., indole, AHR ligands, PSA polyamines
Also in the introduction part of each kidney disease, a brief description of the immune contribution (immune cell types, etc) to the disease should be added.
In paragraph 4.1, the description of the role of IS, PCS, TMAO in CKD should be expanded as these are major microbiota-derived uremic toxins extensively studied in endothelial and tubular damage.
In paragraph 4.3 the following sentence is unclear:
The hallmark of this disease is the deposition of IgA1 deficient in the glomeruli of galactose.
Author Response
We would like to thank for the suggestions of revision, which undoubtedly improved our paper.
This is a very comprehensive review in which Mertowska et al. summarize the present findings on the role and association of gut dysbiosis in different types of CKD.The authors put emphasis on the role of inflammation and dietary factors in the gut-kidney axis. Although this is a clear and very well-written review I do have some comments for improving the manuscript.
In paragraph 3, other bacterial metabolites/components should be described beyond SCFA; e.g., indole, AHR ligands, PSA polyamines
Also in the introduction part of each kidney disease, a brief description of the immune contribution (immune cell types, etc) to the disease should be added.
In paragraph 4.1, the description of the role of IS, PCS, TMAO in CKD should be expanded as these are major microbiota-derived uremic toxins extensively studied in endothelial and tubular damage.
Re: Thank you for the suggestions. All additional descriptions were introduced accordingly. The changes were highlighted in red.
In paragraph 4.3 the following sentence is unclear:
The hallmark of this disease is the deposition of IgA1 deficient in the glomeruli of galactose.
Re: The sentence has been clarified.
Reviewer 2 Report
The authors are to be commended for investigating the existing evidence regarding a link between chronic kidney disease and gut microbiota, deepening immunonological and nutritional aspects. Although it is not a systematic review, the review appears complete and the authors have analyzed the topic in detail. Some minor changes in the structure of the text are need. Moreover, some discussion on other determinants or risk conditions, such as physical activity and smoking, should be add.
Author Response
We would like to thank for the suggestions of revision, which undoubtedly improved our paper.
The authors are to be commended for investigating the existing evidence regarding a link between chronic kidney disease and gut microbiota, deepening immunonological and nutritional aspects. Although it is not a systematic review, the review appears complete and the authors have analyzed the topic in detail. Some minor changes in the structure of the text are need. Moreover, some discussion on other determinants or risk conditions, such as physical activity and smoking, should be add.
Re: Thank you for this comment. We really appreciate your help in this matter. We have added new paragraphs with regard to physical activity and smoking in patients with CKD (p. 18) and proper references were included (no. 144-146). All changes were highlighted in red.